# The Machine Learning landscape of top taggers

Gregor Kasieczka[1⋆], Tilman Plehn[2†], Anja Butter[2], Kyle Cranmer[3], Dipsikha Debnath[4], Barry M. Dillon[5], Malcolm Fairbairn[6], Darius A. Faroughy[5], Wojtek Fedorko[7], Christophe Gay[7], Loukas Gouskos[8], Jernej F. Kamenik[5,9], Patrick T. Komiske[10], Simon Leiss[1], Alison Lister[7], Sebastian Macaluso[3,4], Eric M. Metodiev[10], Liam Moore[11], Ben Nachman[12,13], Karl Nordström[14,15], Jannicke Pearkes[7], Huilin Qu[8], Yannik Rath[16], Marcel Rieger[16], David Shih[4] Jennifer M. Thompson[2], and Sreedevi Varma[6]

**1** Institut für Experimentalphysik, Universität Hamburg, Germany
**2** Institut für Theoretische Physik, Universität Heidelberg, Germany
**3** Center for Cosmology and Particle Physics and Center for Data Science, NYU, USA
**4** NHECT, Dept. of Physics and Astronomy, Rutgers, The State University of NJ, USA
**5** Jozef Stefan Institute, Ljubljana, Slovenia
**6** Theoretical Particle Physics and Cosmology, King's College London, United Kingdom
**7** Department of Physics and Astronomy, The University of British Columbia, Canada
**8** Department of Physics, University of California, Santa Barbara, USA
**9** Faculty of Mathematics and Physics, University of Ljubljana, Ljubljana, Slovenia
**10** Center for Theoretical Physics, MIT, Cambridge, USA
**11** CP3, Universitéxx Catholique de Louvain, Louvain-la-Neuve, Belgium
**12** Physics Division, Lawrence Berkeley National Laboratory, Berkeley, USA
**13** Simons Inst. for the Theory of Computing, University of California, Berkeley, USA
**14** National Institute for Subatomic Physics (NIKHEF), Amsterdam, Netherlands
**15** LPTHE, CNRS & Sorbonne Université, Paris, France
**16** III. Physics Institute A, RWTH Aachen University, Germany

⋆ gregor.kasieczka@uni–hamburg.de, † plehn@uni–heidelberg.de

## Abstract

Based on the established task of identifying boosted, hadronically decaying top quarks, we compare a wide range of modern machine learning approaches. Unlike most established methods they rely on low-level input, for instance calorimeter output. While their network architectures are vastly different, their performance is comparatively similar. In general, we find that these new approaches are extremely powerful and great fun.


---

## Content

# 1 Introduction

Top quarks are, from a theoretical perspective, especially interesting because of their strong interaction with the Higgs boson and the corresponding structure of the renormalization group. Experimentally, they are unique in that they are the only quarks which decay before they hadronize. One of the qualitatively new aspect of LHC physics are the many signal processes which for the first time include phase space regimes with strongly boosted tops. Those are typically analyzed with the help of jet algorithms [1]. Corresponding jet substructure analyses have found their way into many LHC measurements and searches.

Top tagging based on an extension of standard jet algorithms has a long history [2–8]. Standard top taggers used by ATLAS and CMS usually search for kinematic features induced by the top and $W$-boson masses [9–11]. This implies that top tagging is relatively straight-forward, can be described in terms of perturbative quantum field theory, and hence makes an obvious candidate for a benchmark process. An alternative way to tag tops is based on the number of prongs, properly defined as $N$-subjettiness [12]. Combined with the SoftDrop mass variable [13], this defines a particularly economic 2-parameter tagger, but without a guarantee that the full top momentum gets reconstructed.

Based on simple deterministic taggers, the LHC collaborations have established that subjet analyses work and can be controlled in their systematic uncertainties [14–17]. The natural next step are advanced statistical methods [18, 19], including multi-variate analyses [20]. In the same spirit, the natural next question is why we apply highly complex tagging algorithms to a pre-processed set of kinematic observables rather than to actual data. This question becomes especially relevant when we consider the significant conceptual and performance progress in machine learning. Deep learning, or the use of neural networks with many hidden layers,

is the tool which allows us to analyze low-level LHC data without constructing high-level observables. This directly leads us to standard classification tools in contemporary machine learning, for example in image or language recognition.

The goal of this study is to see how well different neutral network setups can classify jets based on calorimeter information. A straightforward way to apply standard machine learning tools to jets is so-called calorimeter images, which we use for our comparison of the different available approaches on an equal footing. Considering calorimeter cells inside a fat jet as pixels defines a sparsely filled image which can be analyzed through standard convolutional networks [21–23]. A set of top taggers defined on the basis of image recognition will be part of our study and will be described in Sec. 3.1 [24, 25]. A second set of taggers is based directly on the 4-momenta of the subjet constituents and will be introduced in Sec. 3.2 [26–28]; recurrent neural networks inspired by language recognition [29] can be grouped into the same category. Finally, there are taggers which are motivated by theoretical considerations like soft and collinear radiation patterns or infrared safety [30–33] which we collect in Sec. 3.3.

While initially it was not clear if any of the machine learning methods applied to top tagging would be able to significantly exceed the performance of the multi-variate tools [24, 26], later studies have consistently showed that we can expect great performance improvement from most modern tools. This turns around the question into which of the tagging approaches have the best performance (also relative to their training effort), and if the leading taggers make use of the same, hence complete set of information. Indeed, we will see that we can consider jet classification based on deep learning at the pure performance level an essentially solved problem. For a systematic experimental application of these tools our focus will be on a new set of questions related to training data, benchmarking, calibration, systematics, etc.

## 2 Data set

The top signal and mixed quark-gluon background jets are produced with using Pythia8 [34] with its default tune for a center-of-mass energy of 14 TeV and ignoring multiple interactions and pile-up. For a simplified detector simulation we use Delphes [35] with the default AT-LAS detector card. This accounts for the curved trajectory of the charged particles, assuming a magnetic field of 2 T and a radius of 1.15 m as well as how the tracking efficiency and momentum smearing changes with $\eta$. The fat jet is then defined through the anti-$k_T$ algorithm [36] in FastJet [37] with $R = 0.8$. We only consider the leading jet in each event and require

$$p_{T,j} = 550 \; .... \; 650 \; \text{GeV} \, . \tag{1}$$

For the signal only, we further require a matched parton-level top to be within $\Delta R = 0.8$, and all top decay partons to be within $\Delta R = 0.8$ of the jet axis as well. No matching is performed for the QCD jets. We also require the jet to have $|\eta_j| < 2$. The constituents are extracted through the Delphes energy-flow algorithm, and the 4-momenta of the leading 200 constituents are stored. For jets with less than 200 constituents we simply add zero-vectors.

Particle information or additional tracking information is not included in this format. For instance, we do not record charge information or the expected displaced vertex from the $b$-decay. Therefore, the quoted performance should not be considered the last word for the LHC. On the other hand, limiting ourselves to essentially calorimeter information allows us to compare many different techniques and tools on an equal footing.

Our public data set consists of 1 million signal and 1 million background jets and can be obtained from the authors upon request [38]. They are divided into three samples: training with 600k signal and background jets each, validation with 200k signal and background jets

each, and testing with 200k signal and 200k background jets. For proper comparison, all algorithms are optimized using the training and validation samples and all results reported are obtained using the test sample. For each algorithm, the classification result for each jet is made available, so we can not only measure the performance of the network, but also test which jets are correctly classified in each approach.

# 3 Taggers

## 3.1 Imaged-based taggers

To evaluate calorimeter information efficiently we can use powerful methods from image recognition. We simply interpret the energy deposition in the pixelled calorimeter over the area of the fat jet as an image and apply convolutional networks to it. These convolutional networks encode the 2-dimensional information which the network would have to learn if we gave it the energy deposition as a 1-dimensional chain of entries. Such 2-dimensional networks are the drivers behind many advances in image recognition outside physics and allow us to benefit from active research in the machine learning community.

If we approximate the calorimeter resolution as $0.04 \times 2.25°$ in rapidity vs azimuthal angle a fat jet with radius parameter $R = 0.8$ can be covered with $40 \times 40$ pixels. Assuming a $p_T$ threshold around 1 GeV, a typical QCD jet will feature around 40 constituents in this jet image [31]. In Fig. 1 we show an individual calorimeter image from a top jet, as well as averaged images of top jets and QCD jets, after some pre-processing. For both, signal and background jets the center of the image is defined by the hardest object. Next, we rotate the second-hardest object to 12 o'clock. Combined with a narrow $p_T$-bin for the jets this second jet develops a preferred distance from the center for the signal but not for the QCD background. Finally, we reflect the third-largest object to the right side of the image, where such a structure is really only visible for the 3-prong top signal. Note that this kind of pre-processing is crucial for visualization, but not necessarily part of a tagger. We will compare two modern deep networks analyzing calorimeter images. A comparison between the established multi-variate taggers and the modestly performing early DeepTop network can be found in Ref. [24].

### 3.1.1 CNN

One standard top tagging method applies a convolutional neural network (CNN) trained on jet images, generated from the list of per-jet constituents of the reference sample [25]. We

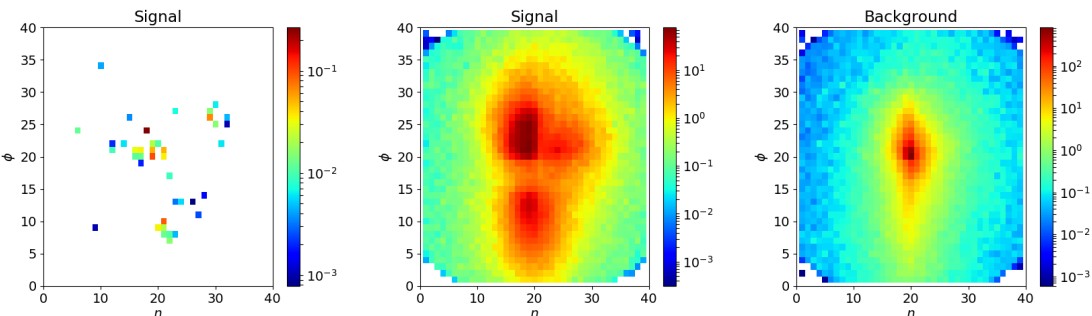

Figure 1: Left: typical single jet image in the rapidity vs azimuthal angle plane for the top signal after pre-processing. Center and right: signal and background images averaged over 10,000 individual images.

perform a specific preprocessing before pixelating the image. First, we center and rotate the jet according to its $p_T$-weighted centroid and principal axis. Then we flip horizontally and vertically so that the maximum intensity is in the upper right quadrant. Finally, we pixelate the image with $p_T$ as the pixel intensity, and normalize it to unit total intensity. Although our original method includes color images, where the track $p_T$'s and neutral $p_T$'s are considered separately, for this dataset we restrict ourselves to gray-scale images.

For the design of our CNN, we take the DeepTop architecture [24] as a starting point, but augment it with more feature maps, hidden units, etc. The complete network architecture is illustrated in Fig. 2.

The CNN is implemented on an NVidia Tesla P100 GPU using PyTorch. For training we use the cross entropy loss function and Adam as the optimizer [39] with a minibatch size of 128. The initial learning rate is $5 \times 10^{-5}$ and is decreased by a factor of 2 every 10 epochs. The CNN is trained for 50 epochs and the epoch with the best validation accuracy is chosen as the final tagger.

### 3.1.2 ResNeXt

The ResNeXt model is another deep CNN using jet images as inputs. The images used in this model are 64×64 pixels in size centered on the jet axis, corresponding to a granularity of 0.025 radians in the $\eta - \phi$ space. The intensity of each pixel is the sum of $p_T$ of all the constituents within the pixel. The CNN architecture is based on the 50-layer ResNeXt architecture [40]. To adapt to the smaller size of the jet images, the number of channels in all the convolutional layers except for the first one is reduced by a factor of 4, and a dropout layer with a keep probability of 0.5 is added after the global pooling. The network in implemented in Apache MXNet [41], and trained from scratch on the top tagging dataset.

### 3.2 4-Vector-based taggers

A problem with the image recognition approaches discussed in Sec. 3.1 arises when we want to include additional information for example from tracking or particle identification. We can always combine different images in one analysis [42], but the significantly different resolution for example of calorimeter and tracker images becomes a serious challenge. For a way out we can follow the experimental approach developed by CMS and ATLAS and use particle flow or similar objects as input to neural network taggers. In our case this means 4-vectors with the energy and the momentum of the jet constituents. The challenge is to define an efficient network setup that either knows or learns the symmetry properties of 4-vectors and replace the notion of 2-dimensional geometric structure included in the convolutional networks

Of the known, leading properties of top jets the 4-vector approach can be used to efficiently extract the number of prongs as well as the number of constituents as a whole. For the latter, it is crucial for the taggers to notice that soft QCD activity is universal and should not provide additional tagging information.

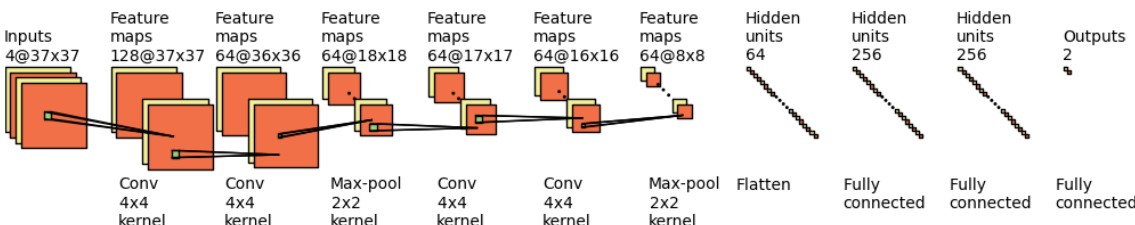

Figure 2: Architecture of the CNN top tagger. Figure from Ref. [25].

### 3.2.1 TopoDNN

If we start with 200 $p_T$-sorted 4-vectors per jet, the arguably simplest deep network architecture is a dense network taking all 800 floating point numbers as a fixed set [27]. Since the mass of individual particle-flow candidates cannot be reliably reconstructed, the TopoDNN tagger uses at most 600 inputs, namely $(p_T, \eta, \phi)$ for each constituent. To improve the training through physics-motivated pre-processing, a longitudinal boost and a rotation in the transverse plane are applied such that the $\eta$ and $\phi$ values of the highest-$p_T$ constituent is centered at $(0, 0)$. The momenta are scaled by a constant factor $1/1700$, chosen ad-hoc because a dynamic quantity such as the constituent $p_T$ can distort the jet mass [22]. A further rotation is applied so that the second highest jet constituent is aligned with the negative $y$-axis, to remove the rotational symmetry of the second prong about the first prong in the jet. This is a proper rotation, not a simple rotation in the $\eta$-$\phi$ plane, and thus preserves the jet mass (but can distort quantities like $N$-subjettiness [12, 43]).

The TopoDNN tagger presented here has a similar architecture as a tagger with the same name used by the ATLAS collaboration [44]. The architecture of this TopoDNN tagger was optimized for a dataset of high $p_T$ (450 to 2400 GeV) $R = 1.0$ trimmed jets. Its hyper-parameters are the number of constituents considered as inputs, the number of hidden layers, the number of nodes per layer, and the activation functions for each layer. For the top dataset of this study we find that 30 constituents saturate the network performance. The ATLAS tagger uses only 10 jet constituents, but the inputs are topoclusters [45] and not individual particles. The remaining ATLAS TopoDNN architecture is not altered.

Individual particle-flow candidates in the experiment are also not individual particles, but there is a closer correspondence. For this reason, the TopoDNN tagger performance presented here is not directly comparable to the results presented in Ref. [44].

### 3.2.2 Multi-Body N-Subjettiness

The multi-body phase space tagger is based on the proposal in Ref. [46] to use a basis of $N$-subjettiness variables [12] spanning an $m$-body phase space to teach a dense neural network to separate background and signal using a minimal set of input variables. A setup for the specific purpose of top tagging was introduced in Ref. [33]. To generate the input for our network we first convert the event to HEPMC and use Rivet [47] to evaluate $N$-subjettiness variables by using the FastJet [37] contrib plug-in library [12, 48]. The input variables spanning the

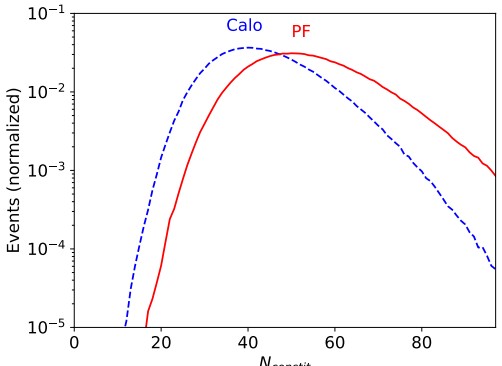
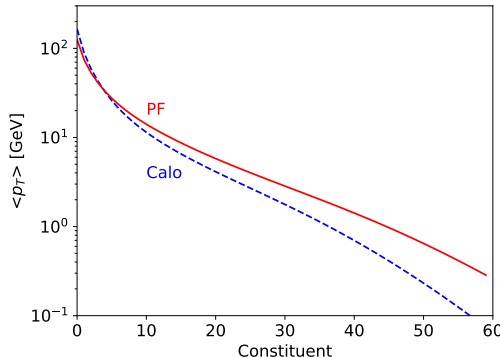

Figure 3: Number of constituents (left) and mean of the transverse momentum (right) of the ranked constituents of a typical top jet. We show calorimeter entries as well as particle flow constituents after Delphes.

$m$-body phase space are then given by:

$$\left\{ \tau_1^{(0.5)}, \tau_1^{(1)}, \tau_1^{(2)}, \tau_2^{(0.5)}, \tau_2^{(1)}, \tau_2^{(2)}, \ldots, \tau_{m-2}^{(0.5)}, \tau_{m-2}^{(1)}, \tau_{m-2}^{(2)}, \tau_{m-1}^{(1)}, \tau_{m-1}^{(2)} \right\}, \tag{2}$$

where

$$\tau_N^{(\beta)} = \frac{1}{p_{T,J}} \sum_{i \in J} p_{T,i} \min\left\{ R_{1i}^{\beta}, R_{2i}^{\beta}, \ldots, R_{Ni}^{\beta} \right\}, \tag{3}$$

and $R_{ni}$ is the distance in the $\eta - \phi$ plane of the jet constituent $i$ to the axis $n$. We choose the $N$ jet axes using the $k_T$ algorithm [49] with $E$-scheme recombination, specifically $N = 6, 8$ for our explicit comparison. This input has the advantage that it is theoretically sound and IR-safe, while it can be easily understood as a set of correlators between 4-momenta combining the number of prongs in the jet with additional momentum information.

The machine learning setup is a dense neural network implemented in TensorFlow [50] using these input variables. We also add the jet mass and jet $p_T$ as input variables to allow the network to learn physical scales. The network consists of four fully connected hidden layers, the first two with 200 nodes and a dropout regularization of 0.2, and the last two with 50 nodes and a dropout regularization of 0.1. The output layer consists of two nodes. We use a ReLu activation function throughout and minimize the cross-entropy using Adam optimization [39].

### 3.2.3 TreeNiN

In this method, a tree neural network (TreeNN) is trained on jet trees. The TreeNN provides a jet embedding, which maps a set of 4-momenta into a vector of fixed size and can be trained together with a successive network used for classification or regression [29]. Jet constituents of the reference sample are reclustered to form binary trees, and the topology is determined by the clustering algorithm, *i.e.* $k_T$, anti-$k_T$ or Cambridge/Aachen. For this paper, we choose the $k_T$ clustering algorithm, and 7 features for the nodes: $|p|$, $\eta$, $\phi$, $E$, $E/E_{\text{jet}}$, $p_T$ and $\theta$. We scale each feature with the Scikit-learn preprocessing method RobustScaler, which is robust to outliers [51].

To speed up the training, a special batching is implemented in Ref. [29]. Jets are reorganized by levels, *e.g.* the root node of each tree in the batch is added at level zero, their children at level one, etc. Each level is restructured such that all internal nodes come first, followed by outer nodes (leaves), and zero padding is applied when necessary. We developed a PyTorch implementation to provide GPU acceleration.

We introduce in Ref. [52] a Network-in-Network generalization of the simple TreeNN architecture proposed in Ref. [29], where we add fully connected layers at each node of the binary tree before moving forward to the next level. We refer to this model as TreeNiN. In particular, we add 2 NiN layers with ReLU activations. Also, we split weights between internal nodes and leaves, both for the NiN layers and for the initial embedding of the 7 input features of each node. Finally, we introduce two sets of independent weights for the NiN layers of the left and right children of the root node. The code is publicly accessible on GitHub [53]. Training is performed over 40 epochs with a minibatch of 128 and a learning rate of $2 \times 10^{-3}$ decayed by a factor of 0.9 after every epoch, using the cross entropy loss function and the Adam optimizer [39].

### 3.2.4 P-CNN

The particle-level convolutional neural network (P-CNN), used in the CMS particle-based DNN tagger [54], is a customized 1-dimensional CNN for boosted jet tagging. Each input jet is represented as a sequence of constituents with a fixed length of 100, organized in descending order of $p_T$. The sequence is padded with zeros if a jet has less than 100 constituents. If

a jet contains more than 100 constituents, the extra constituents are discarded. For each constituent, seven input features are computed from the 4-momenta of the constituent and used as inputs to the network: $\log p_T^i$, $\log E^i$, $\log(p_T^i/p_T^{\text{jet}})$, $\log(E^i/E^{\text{jet}})$, $\Delta\eta^i$, $\Delta\phi^i$ and $\Delta R^i$. Angular distances are measured with respect to the jet axis. The use of these transformed features instead of the raw 4-momenta was found to lead to slightly improved performance.

The P-CNN used in this paper follows the same architecture as the CMS particle-based DNN tagger. However, the top tagging dataset in this paper contains only kinematic information of the particles, while the CMS particle-based DNN tagger uses particle tracks and secondary vertices in addition. Therefore, the network components related to them are removed for this study. The P-CNN is similar to the ResNet model [55] for image recognition, but only uses a 1-dimensional convolution instead of 2-dimensional convolutions. One of the features that distinguishes the ResNet architecture from other CNNs is that it includes skip connections between layers. The number of convolutional layers is 14, all with a kernel size of 3. The number of channels for the 1-dimensional convolutional layers ranges from 32 to 128. The outputs of the convolutional layers undergo a global pooling, followed by a fully-connected layer of 512 units and a dropout layer with a keep rate of 0.5, before yielding the final prediction. The network is implemented in Apache MXNet [41] and trained with the Adam optimizer [39].

### 3.2.5 ParticleNet

Similar to the Particle Flow Network, the ParticleNet [56] is also built on the point cloud representation of jets, where each jet is treated as an unordered set of constituents. The input features for each constituent are the same as that in the P-CNN model. The ParticleNet first constructs a graph for each jet, with the constituents as the vertices. The edges of the graph are then initialized by connecting each constituent to its $k$ nearest-neighbor constituents based in $\eta - \phi$ space. The EdgeConv operation [57] is then applied on the graph to transform and aggregate information from the nearby constituents at each vertex, analogous to how regular convolution operates on square patches of images. The weights of the EdgeConv operator are shared among all constituents in the graph, therefore preserving the permutation invariance property of the constituents in a jet. The EdgeConv operations can be stacked to form a deep graph convolutional network.

The ParticleNet relies on the dynamic graph convolution approach of Ref. [57] and further extends it. The ParticleNet consists of three stages of EdgeConv operations, with three Edge-Conv layers and a shortcut connection [58] at each stage. Between the stages, the jet graph is dynamically updated by redefining the edges based on the distances in the new feature space generated by the EdgeConv operations. The number of nearest neighbors, $k$, is also varied in each stage. The three stages of EdgeConv are followed by a global pooling over all constituents, and then two fully connected layers. The details of the ParticleNet architecture can be found in [56]. It is implemented in Apache MXNet [41] and trained with Adam [39].

## 3.3 Theory-inspired taggers

Going beyond a relatively straightforward analysis of 4-vectors we can build networks specifically for subjet analyses and include as much of our physics knowledge as possible. The motivation for this is two-fold: building this information into the network should save training time, and it should allow us to test what kind of physics information the network relies on.

At the level of these 4-vectors the main difference between top jets and massless QCD jets is two mass drops [2,10], which appear after we combine 4-vectors based on soft and collinear proximity. While it is possible for taggers to learn Lorentz boosts and the Minkowski metric, it might be more efficient to give this information as part of the tagger input and architectures.

In addition, any jet analysis tool should give stable results in the presence of additional soft or collinear splittings. From theory we know that smooth limits from very soft or collinear splittings to no splitting have to exist, a property usually referred to as infrared safety. If we replace the relatively large pixels of calorimeter images with particle-level observables it is not clear how IR-safe a top tagging output really is [59]. This is not only a theoretical problem which arises when we for example want to compare rate measurements with QCD predictions, a lack of IR-safety will also make it hard to train or benchmark taggers on Monte Carlo simulations and to extraction tagging efficiencies using Monte Carlo input [60].

### 3.3.1 Lorentz Boost Network

The Lorentz Boost Network (LBN) is designed to autonomously extract a comprehensive set of physics-motivated features given only low-level variables in the form of constituent 4-vectors [28]. These engineered features can be utilized in a subsequent neural network to solve a specific physics task. The resulting two-stage architecture is trained jointly so that extracted feature characteristics are adjusted during training to serve the minimization of the objective function by means of back-propagation.

The general approach of the LBN is to reconstruct parent particles from 4-vectors of their decay products and to exploit their properties in appropriate rest frames. Its architecture is comprised of three layers. First, the input vectors are combined into $2 \times M$ intermediate vectors through linear combinations. Corresponding linear coefficients are trainable and constrained to positive numbers to prevent constructing vectors with unphysical implications, such as $E < 0$ or $E < m$. In the subsequent layer, half of these intermediate vectors are treated as constituents, whereas the other half are considered rest frames. Via Lorentz transformation the constituents are boosted into the rest frames in a pairwise approach, *i.e.* the $m^\text{th}$ constituent is boosted into the $m^\text{th}$ rest frame. In the last layer, $F$ features are extracted from the obtained $M$ boosted constituents by employing a set of generic feature mappings. The autonomy of the LBN lies in its freedom to construct arbitrary boosted particles through trainable particle and rest frame combinations, and to consequently access and provide underlying characteristics which are otherwise distorted by relativistic kinematics.

The order of input vectors is adjusted before being fed to the LBN. The method utilizes linearized clustering histories as preferred by the anti-$k_T$ algorithm [36], implemented in Fastjet [37]. First, the jet constituents are reclustered with $\Delta R = 0.2$, and the resulting subjets are ordered by $p_T$. Per subjet, another clustering with $\Delta R = 0.2$ is performed and the order of constituents is inferred from the anti-$k_T$ clustering history. In combination, this approach yields a consistent order of the initial jet constituents.

The best training results for this study are obtained for $M = 50$ and six generic feature mappings: $E$, $m$, $p_T$, $\phi$, and $\eta$ of all boosted constituents, and the cosine of the angle between momentum vectors of all pairs of boosted constituents, in total $F = 5M + (M^2 - M)/2$ features. Batch normalization with floating averages during training is employed after the feature extraction layer [61]. This subsequent neural network incorporates four hidden layers, involving 1024, 512, 256, and 128 exponential linear (ELU) units, respectively. Generalization and overtraining suppression are enforced via L2 regularization with a factor of $10^{-4}$. The Adam optimizer is utilized for minimizing the binary cross-entropy loss [39], configured with an initial learning rate of $10^{-3}$ for a batch size of 1024. The training procedure converges after around 5000 batch iterations.

### 3.3.2 Lorentz Layer

Switching from image recognition to a setup based on 4-momenta we can take inspiration from graph convolutional networks. Also used for the ParticleNet they allow us to analyze

sparsely filled images in terms of objects and a free distance measure. While usually the most appropriate distance measure needs to be determined from data, fundamental physics tells us that the relevant distance for jet physics is the scalar product of two 4-vectors. This scalar product will be especially effective in searching for heavy masses in a jet clustering history when we evaluate it for combinations of final-state objects.

The input to the Lorentz Layer (LoLa) network [31] are sets of 4-vectors $k_{\mu,i}$ with $i = 1, ..., N$. As a first step we apply a combination layer to define linear combinations of the 4-vectors,

$$k_{\mu,i} \xrightarrow{\text{CoLa}} \widetilde{k}_{\mu,j} = k_{\mu,i}\, C_{ij} \quad \text{with} \quad C = \begin{pmatrix} 1 & \cdots & 0 & C_{1,N+1} & \cdots & C_{1,M} \\ \vdots & \ddots & \vdots & \vdots & & \vdots \\ 0 & \cdots & 1 & C_{N,N+1} & \cdots & C_{N,M} \end{pmatrix}. \tag{4}$$

Here we use $N = 60$ and $M = 90$. In a second step we transform all 4-vectors into measurement-motivated objects,

$$\widetilde{k}_j \xrightarrow{\text{LoLa}} \hat{k}_j = \begin{pmatrix} m^2(\widetilde{k}_j) \\ p_T(\widetilde{k}_j) \\ w_{jm}^{(E)}\, E(\widetilde{k}_m) \\ w_{jm}^{(p_T)}\, p_T(\widetilde{k}_m) \\ w_{jm}^{(m^2)}\, m^2(\widetilde{k}_m) \\ w_{jm}^{(d)}\, d_{jm}^2 \end{pmatrix}, \tag{5}$$

where $d_{jm}^2$ is the Minkowski distance between two 4-momenta $\widetilde{k}_j$ and $\widetilde{k}_m$, and we either sum or minimize over the internal indices. One copy of the sum and five copies of the minimum term are used. Just for amusement we have checked with what precision the network can learn the Minkowski metric from top vs QCD jet data [31]. The LoLa network setup is then straightforward, three fully connected hidden layers with 100, 50, and 2 nodes, and using the Adam optimizer [39]

### 3.3.3 Latent Dirichlet Allocation

Latent Dirichlet Allocation (LDA) is a widely used unsupervised learning technique used in generative modelling for collections of text documents [62]. It can also uncover the latent thematic structures in jets or events by searching for co-occurrence patterns in high-level sub-structure features [63]. Once the training is performed the result is a set of learned themes, *i.e.* probability distributions over the substructure observable space. For the top tagger, we use a two-theme LDA model, aimed at separate themes describing the signal and background. We then use these distributions to infer theme proportions and tag jets. Both the training and inference are performed with the Gensim software [64].

Before training, we pre-process the jets and map them to a representation suitable for the Gensim software. After clustering the jets with the Cambridge/Aachen algorithm with a large cone radius, each jet is iteratively unclustered without discarding any of the branches in the unclustering history. At each step of unclustering we compute a set of substructure observables from the parent subjet and the two daughter subjets, namely the subjet mass, the mass drop, the mass ratio of the daughters, the angular separation of the daughters, and the helicity angle of the parent subjet in the rest frame of the heaviest daughter subjet. These five quantities are collated into a 5-dimensional feature vector for each node of the tree until the unclustering is done. This represents each jet as a list of 5-vector substructure features. To improve the top tagging we also include complementary information, namely the $N$-subjettiness observables, $(\tau_3/\tau_2, \tau_3/\tau_1, \tau_2/\tau_1)$. All eight observables are binned and mapped to a vocabulary that is used to transform the jet into a training document for the two-theme LDA.

LDA is generally used as an unsupervised learning technique. For this study we use, for the sake of comparison, the LDA algorithm in a supervised learning mode. Regardless of whether the training is performed in a supervised or unsupervised manner, once it is complete it is straightforward to study what has been learned by the LDA algorithm by inspecting the theme distributions. For instance, in Ref. [63] the uncovered latent themes from the data are plotted and one can easily distinguish QCD-like features in one theme and top-like features in the other.

### 3.3.4 Energy Flow Polynomials

Energy Flow Polynomials (EFPs) [30] are a collection of observables designed to form a linear basis of all infrared- and collinear- (IRC) safe observables, building upon a rich literature of energy correlators [65–67]. The EFPs naturally enable linear methods to be applied to collider physics problems, where the simplicity and convexity of linear models is highly desirable.

EFPs are energy correlators whose angular structures are in correspondence with non-isomorphic multigraphs. Specifically, they are defined using the transverse momentum fractions, $z_i = p_{T,i} / \sum_j p_{T,j}$, of the $M$ constituents as well as their pairwise rapidity-azimuth distances, $\theta_{ij} = ((y_i - y_j)^2 + (\phi_i - \phi_j)^2)^{\beta/2}$, without any additional preprocessing:

$$\text{EFP}_G = \sum_{i_1=1}^{M} \cdots \sum_{i_N=1}^{M} z_{i_1} \cdots z_{i_N} \prod_{(k,\ell) \in G} \theta_{i_k i_\ell}, \tag{6}$$

where $G$ is a given multigraph, $N$ is the number of vertices in $G$, and $(k, \ell)$ is an edge connecting vertices $k$ and $\ell$ in $G$. The EFP-multigraph correspondence yields simple visual rules for translating multigraphs to observables: vertices contribute an energy factor and edges contribute an angular factor. Beyond this, the multigraphs provide a natural organization of the EFP basis when truncating in the number of edges (or angular factors) $d$, with exactly 1000 EFPs with $d \leq 7$.

For the top tagging EFP model, all $d \leq 7$ EFPs computable in $\mathcal{O}(M^3)$ or faster are used (995 observables total) with angular exponent $\beta = 1/2$. Linear classification was performed with Fisher's Linear Discriminant [68] using Scikit-learn [51]. Implementations of the EFPs are available in the EnergyFlow package [69].

### 3.3.5 Energy Flow Networks

An Energy Flow Network (EFN) [32] is an architecture built around a general decomposition of IRC-safe observables that manifestly respects the variable-length and permutation-invariance symmetries of observables. Encoding the proper symmetries of particle collisions in an architecture results in a natural way of processing the collider data.

Any IRC-safe observable can be approximated arbitrarily well as

$$\text{EFN} = F\left( \sum_{i=1}^{M} z_i \Phi(y_i, \phi_i) \right), \tag{7}$$

where $\Phi : \mathbb{R}^2 \to \mathbb{R}^\ell$ is a per-constituent mapping that embeds the locations of the constituents in a latent space of dimension $\ell$. Constituents are summed over in this latent space to obtain an event representation, which is then mapped by $F$ to the target space of interest.

The EFN architecture parametrizes the functions $\Phi$ and $F$ with neural networks, with specific implementation details of the top tagging EFN architecture given in the Particle Flow Network section. For both the top tagging EFNs and PFNs, input constituents are translated to the origin of the rapidity-azimuth plane according to the $p_T$-weighted centroid, rotated in

this plane to consistently align the principal axis of the energy flow, and reflected to locate the highest $p_T$ quadrant in a consistent location. A fascinating aspect of the decomposition in Eq. (7) is that the learned latent space can be examined both quantitatively and visually, as the rapidity-azimuth plane is two dimensional and can be viewed as an image. Figure 4 shows a visualization of the top tagging EFN, which learns a dynamic pixelization of the space.

### 3.3.6 Particle Flow Networks

Particle Flow Networks (PFNs) [32] generalize the EFNs beyond IRC safety. In doing so, they make direct contact with machine learning models on learning from point clouds, in particular the Deep Sets framework [70]. This identification of point clouds as the machine learning data structure with intrinsic properties most similar to collider data provides a new avenue of exploration.

In the collider physics language, the key idea is that any observable can be approximated arbitrarily well as

$$\text{PFN} = F\left(\sum_{i=1}^{M} \Phi(p_i)\right),\tag{8}$$

where $p_i$ contains per-particle information, such as momentum, charge, or particle type. Similar to the EFN case, $\Phi$ maps from the particle feature space into a latent space and $F$ maps from the latent space to the target space. The per-particle features provided to the network can be varied to study their information content. Only constituent 4-momentum information is used here. Exploring the importance of particle-type information for top tagging is an interesting direction for future studies.

The PFN architecture parameterizes the functions $\Phi$ and $F$ with neural networks. For the top tagging EFN and PFN models, $\Phi$ and $F$ are parameterized with simple three-layer neural networks of $(100, 100, 256)$ and $(100, 100, 100)$ nodes in each layer, respectively, corresponding to a latent space dimension of $\ell = 256$. A ReLU activation is used for each dense layer

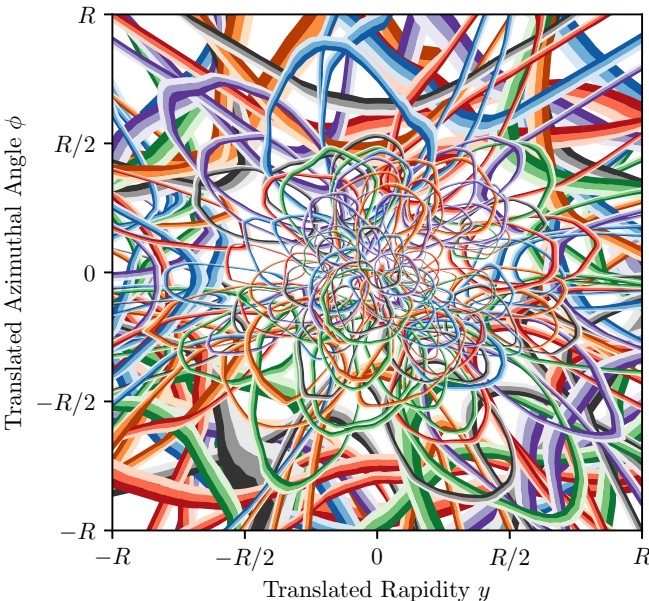

Figure 4: Visualization of the trained top tagging EFN. Each contour corresponds to a filter, which represents the learned local latent space. The smaller filters probe the core of the jet and larger filters the periphery. Figure from Ref. [32].

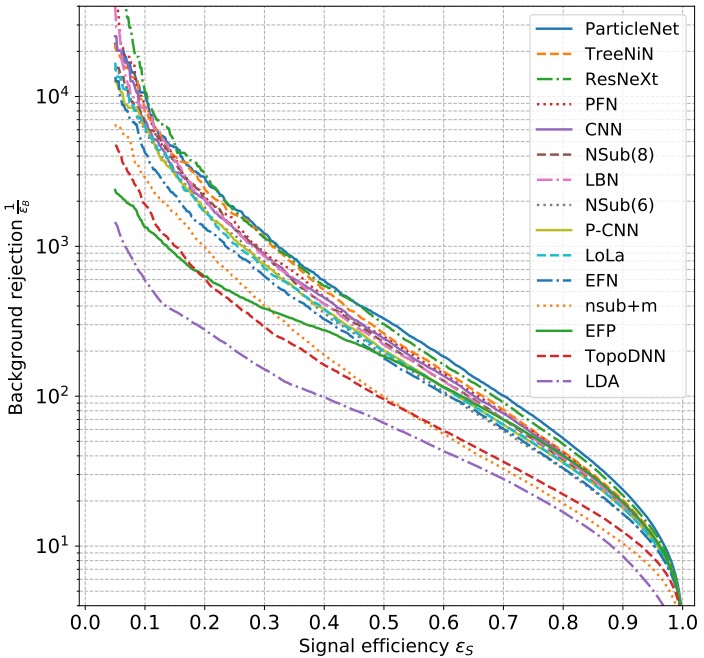

Figure 5: ROC curves for all algorithms evaluated on the same test sample, shown as the AUC ensemble median of multiple trainings. More precise numbers as well as uncertainty bands given by the ensemble analysis are given in Tab. 1.

along with He-uniform parameter initialization [71], and a two-node classifier output is used with SoftMax activation, trained with Keras [72] and TensorFlow [50]. Implementations of the EFNs and PFNs are available in the EnergyFlow package [69].

## 4 Comparison

To assess the performance of different algorithms we first look at the individual ROC curves over the full range of top jet signal efficiencies. They are shown in Fig. 5, compared to a simple tagger based on $N$-subjettiness [12] and jet mass. We see how, with few exceptions, the different taggers define similar shapes in the signal efficiency vs background rejection plane.

Given that observation we can instead analyze three single-number performance metrics for classification tasks. First, we compute the *area under the ROC curve* shown in Fig. 5. It is bounded to be between 0 and 1, and stronger classification corresponds to values larger than 0.5 at a chosen working point. Next, the *accuracy* is defined as the fraction of correctly classified jets. Finally, for a typical analysis application the *rejection power* at a realistic working point is most relevant. We choose the background rejection at a signal efficiency of 30%.

All three figures of merit are shown in Tab. 1. Most approaches achieve an AUC of approximately 0.98 with the strongest performance from the 4-vector-based ParticleNet, followed by the image-based ResNeXt, the 4-vector-based TreeNiN, and the theory-inspired Particle Flow Network. These approaches also reach the highest accuracy and background rejection at fixed signal efficiency. A typical accuracy is 93%, and the quoted differences between the taggers

Table 1: Single-number performance metrics for all algorithms evaluated on the test sample. We quote the area under the ROC curve (AUC), the accuracy, and the background rejection at a signal efficiency of 30%. For the background rejection we also show the mean and median from an ensemble tagger setup. The number of trainable parameters of the model is given as well. Performance metrics for the GoaT meta-tagger are based on a subset of events.

| | AUC | Acc | $1/\epsilon_B$ ($\epsilon_S = 0.3$) | | | #Param |
| | | | single | mean | median | |
|---|---|---|---|---|---|---|
| CNN [25] | 0.981 | 0.930 | 914±14 | 995±15 | 975±18 | 610k |
| ResNeXt [40] | 0.984 | 0.936 | 1122±47 | 1270±28 | 1286±31 | 1.46M |
| TopoDNN [27] | 0.972 | 0.916 | 295±5 | 382± 5 | 378 ± 8 | 59k |
| Multi-body $N$-subjettiness 6 [33] | 0.979 | 0.922 | 792±18 | 798±12 | 808±13 | 57k |
| Multi-body $N$-subjettiness 8 [33] | 0.981 | 0.929 | 867±15 | 918±20 | 926±18 | 58k |
| TreeNiN [52] | 0.982 | 0.933 | 1025±11 | 1202±23 | 1188±24 | 34k |
| P-CNN | 0.980 | 0.930 | 732±24 | 845±13 | 834±14 | 348k |
| ParticleNet [56] | 0.985 | 0.938 | 1298±46 | 1412±45 | 1393±41 | 498k |
| LBN [28] | 0.981 | 0.931 | 836±17 | 859±67 | 966±20 | 705k |
| LoLa [31] | 0.980 | 0.929 | 722±17 | 768±11 | 765±11 | 127k |
| LDA [63] | 0.955 | 0.892 | 151±0.4 | 151.5±0.5 | 151.7±0.4 | 184k |
| Energy Flow Polynomials [30] | 0.980 | 0.932 | 384 | | | 1k |
| Energy Flow Network [32] | 0.979 | 0.927 | 633±31 | 729±13 | 726±11 | 82k |
| Particle Flow Network [32] | 0.982 | 0.932 | 891±18 | 1063±21 | 1052±29 | 82k |
| GoaT | 0.985 | 0.939 | 1368±140 | | 1549±208 | 35k |

are unlikely to define a clear experimental preference.

Instead of extracting these performance measures from single models we can use ensembles. For this purpose we train nine models for each tagger and define 84 ensemble taggers, each time combining six of them. They allow us to evaluate the spread of the ensemble taggers and define mean-of-ensemble and median-of-ensemble results. We find that ensembles leads to a 5 ... 15% improvement in performance, depending on the algorithm. For the uncertainty estimate of the background rejection we remove the outliers. In Tab. 1 we see that the background rejection varies from around 1/600 to better than 1/1000. For the ensemble tagger the ParticleNet, ResNeXt, TreeNiN, and PFN approaches again lead to the best results. Phrased in terms of the improvement in the signal-to-background ratio they give factors $\epsilon_S/\epsilon_B > 300$, vastly exceeding the current top tagging performance in ATLAS and CMS.

Altogether, in Fig. 5 and Tab. 1 we see that some of the physics-motivated setups remain competitive with the technically much more advanced ResNeXt and ParticleNet networks. This suggests that even for a straightforward task like top tagging in fat jets we can develop efficient physics-specific tools. While their performance does not quite match the state-of-the-art standard networks, it is close enough to test both approaches on key requirements in particle physics, like treatment of uncertainties, stability with respect to detector effects, etc.

The obvious question in any deep-learning analysis is if the tagger captures all relevant information. At this point we have checked that including full or partial information on the event-level kinematics of the fat jets in the event sample has no visible impact on our quoted performance metrics. We can then test how correlated the classifier output of the different taggers are, leading to the pair-wise correlations for a subset of classifier outputs shown in Fig. 6. The correlation matrix is given in Tab. 2. As expected from strong classifier performances, most jets are clustered in the bottom left and top right corners, corresponding to identification as background and signal, respectively. The largest spread is observed for correlations with

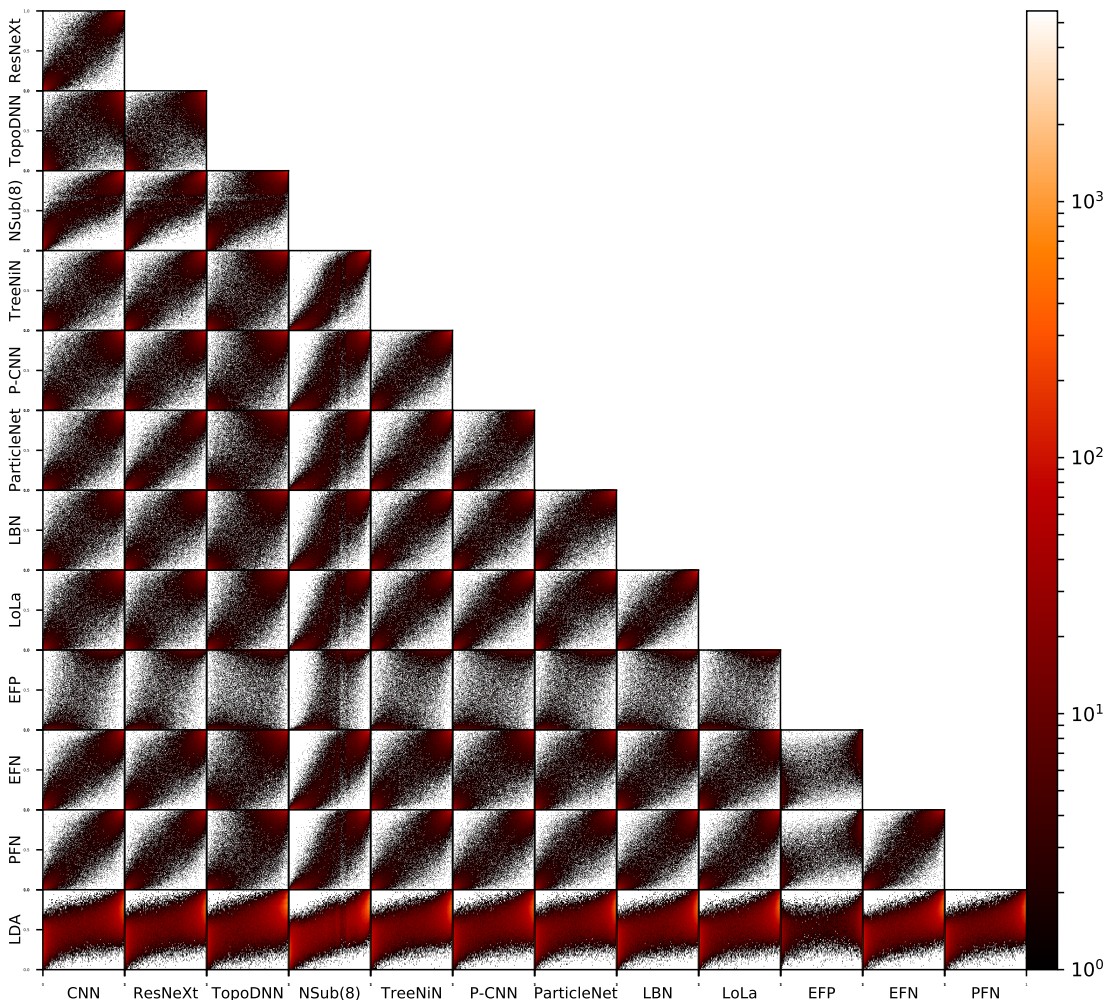

Figure 6: Pairwise distributions of classifier outputs, each in the range 0 ... 1 from pure QCD to pure top; the lower left corners include correctly identified QCD jets, while the upper right corners are correctly identified top jets. LDA outputs are rescaled by a factor of $\approx 4$ to have the same range as other classifiers.

the EFP. Even the two strongest individual classifier outputs with relatively little physics input — ResNeXt and ParticleNet — are not perfectly correlated.

Given this limited correlation, we investigate whether a meta-tagger might improve performance. Note that this GoaT (Greatest of all Taggers) meta-tagger should not be viewed as a potential analysis tool, but rather as a benchmark of how much unused information is still available in correlations. It is implemented as a fully connected network with 5 layers containing 100-100-100-20-2 nodes. All activation functions are ReLu, apart from the final layer's SoftMax. Training is performed with the Adam [39] optimizer with an initial learning rate of 0.001 and binary cross-entropy loss. We train for up to 50 epochs, but terminate if there is no improvement in the validation loss for two consecutive epochs, so a typical training ends after 5 epochs. The training data is provided by individual tagger output on the previous test sample and split intro three subsets: GoaT-training (160k events), GoaT-testing (160k events) and GoaT-validation (80k events). We repeat training/testing nine times, re-shuffling the events

Table 2: Correlation coefficients from the combined GoaT analyses

|  | CNN | ResNeXt | TopoDNN | NSub(8) | TreeNiN | P-CNN | ParticleNet | LBN | LoLa | EFP | EFN | PFN | LDA |
|---|---|---|---|---|---|---|---|---|---|---|---|---|---|
| CNN | 1 | .983 | .955 | .978 | .977 | .973 | .979 | .976 | .974 | .973 | .981 | .984 | .784 |
| ResNeXt | .983 | 1 | .953 | .977 | .98 | .975 | .985 | .977 | .974 | .975 | .976 | .983 | .782 |
| TopoDNN | .955 | .953 | 1 | .962 | .958 | .962 | .953 | .961 | .97 | .945 | .955 | .961 | .777 |
| NSub(8) | .978 | .977 | .962 | 1 | .982 | .975 | .975 | .977 | .977 | .964 | .979 | .977 | .802 |
| TreeNiN | .977 | .98 | .958 | .982 | 1 | .98 | .982 | .982 | .981 | .968 | .973 | .981 | .786 |
| P-CNN | .973 | .975 | .962 | .975 | .98 | 1 | .978 | .98 | .984 | .964 | .968 | .98 | .781 |
| ParticleNet | .979 | .985 | .953 | .975 | .982 | .978 | 1 | .978 | .977 | .97 | .972 | .981 | .778 |
| LBN | .976 | .977 | .961 | .977 | .982 | .98 | .978 | 1 | .984 | .968 | .972 | .98 | .784 |
| LoLa | .974 | .974 | .97 | .977 | .981 | .984 | .977 | .984 | 1 | .968 | .971 | .981 | .782 |
| EFP | .973 | .975 | .945 | .964 | .968 | .964 | .97 | .968 | .968 | 1 | .968 | .977 | .764 |
| EFN | .981 | .976 | .955 | .979 | .973 | .968 | .972 | .972 | .971 | .968 | 1 | .981 | .792 |
| PFN | .984 | .983 | .961 | .977 | .981 | .98 | .981 | .98 | .981 | .977 | .981 | 1 | .784 |
| LDA | .784 | .782 | .777 | .802 | .786 | .781 | .778 | .784 | .782 | .764 | .792 | .784 | 1 |

randomly between the three subsets for each repetition. The standard deviation of these nine repetitions is reported as uncertainty for GoaT taggers in Tab. 1. We show two GoaT versions, one using a single output value per tagger as input (15 inputs), and one using all values per tagger as input (135 inputs). All described taggers are used as input except LDA as it did not improve performance. We see that the GoaT combination improves the best individual tagger by more than 10% in background rejection, providing us with a realistic estimate of the kind of improvement we can still expect for deep-learning top taggers.

In spite of the fact that our study gives some definite answers concerning deep learning for simple jet classification at the LHC, a few questions remain open: first, we use jets in a relatively narrow and specific $p_T$-slice. Future efforts could explore softer jets, where the decay products are not necessarily inside one fat jet; higher $p_T$, where detector resolution effects become crucial; and wider $p_T$ windows, where stability of taggers becomes relevant. The samples also use a simple detector simulation and do not contain effects from underlying event and pile-up.

Second, our analysis essentially only includes calorimeter information as input. Additional information exists in the distribution of tracks of charged particles and especially the displaced secondary vertices from decays of $B$-hadrons. How easily this information can be included might well depend on details of the network architecture.

Third, when training machine learning classifiers on simulated events and evaluating them on data, there exists a number of systematic uncertainties that need to be considered. Typical examples are jet calibration, MC generator modeling, and IR-safety when using theory predictions. Understanding these issues will be a crucial next step. Possibilities are to include uncertainties in the training or to train on data in a weakly supervised or unsupervised fashion [73–80].

Finally, we neglect questions such as computational complexity, evaluation time and memory footprint. These will be important considerations, especially once we want to include deep networks in the trigger. A related questions will be how many events we need to saturate the performance of a given algorithm.

# 5  Conclusion

Because it is experimentally and theoretically well defined, top tagging is a prime benchmark candidate to determine what we can expect from modern machine learning methods in classification tasks relevant for the LHC. We have shown how different neural network architectures and different representations of the data distinguish hadronically decaying top quarks from a background of light quark or gluon jets.

We have compared three different classes of deep learning taggers: image-based networks, 4-vector-based networks, and taggers relying on additional considerations from relativistic kinematics or theory expectations. We find that each of these approaches provide competitive taggers with comparable performance, making it clear that there is no golden deep network architecture. Simple tagging performance will not allow us to identify the kind of network architectures we want to use for jet classification tasks at the LHC. Instead, we need to investigate open questions like versatility and stability in the specific LHC environment with the specific requirements of the particle physics community for example related to calibration and uncertainty estimates.

This result is a crucial step in establishing deep learning at the LHC. Clearly, jet classification using deep networks working on low-level observables is the logical next step in subjet analysis to be fully exploited by ATLAS and CMS. On the other hand, there exist a range of open questions related to training, calibration, and uncertainties. They are specific to LHC physics and might well require particle physics to move beyond applying standard tagging approaches and in return contribute to the field of deep learning. Given that we have now understood that different network setups can achieve very similar performance for mostly calorimeter information, it is time to tackle these open questions.

## Acknowledgments

First and foremost we would like to thank Jesse Thaler for his insights and his consistent and energizing support, and Michael Russell for setting up the event samples for this comparison. We also would like to thank Sebastian Macaluso and Simon Leiss for preparing the plots and tables in this paper. In addition, we would like to thank all organizers of the ML4Jets workshops in Berkeley and at FNAL for triggering and for supporting this study. Finally, we are also grateful to Michel Luchmann for providing us with Fig. 1. PTK and EMM were supported by the Office of Nuclear Physics of the U.S. Department of Energy (DOE) under grant DE-SC-0011090 and by the DOE Office of High Energy Physics under grant DE-SC-0012567, with cloud computing resources provided through a Microsoft Azure for Research Award. The work of BN is supported by the DOE under contract DE-AC02-05CH11231. KC and SM are supported from The Moore-Sloan Data Science Environment, and NSF OAC-1836650 and NSF ACI-1450310 grants. SM gratefully acknowledges the support of NVIDIA Corporation with the donation of a Titan V GPU used for this project.

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
