# Peer review of "The Machine Learning Landscape of Top Taggers"

_SciPost Physics, doi:SciPost Phys. 7, 014 (2019)_

## Round 2 · Referee Report · Anonymous (Referee 1) · 2019-5-13

Strengths

1 - This paper contains a (needed) comprehensive overview on the performance of relevant machine learning techniques in tagging boosted top quark decays.
2 - The presentation of its findings includes innovative and original approaches to the attempt of understanding the tagger outputs and their correlations. In particular the explicit presentation of the correlation matrix is not very common but highly appreciated.
3 - The rather detailed description of the three different machine learning approaches respectively based on images, jet constituent kinematics (four-vectors) and expectations from physics models (motivated by theory) is very useful for the non-expert readers (see also comments on weaknesses below)
4 - The evaluation of the performances in the limited phase space and with the (selective) Monte Carlo model used to generate the signal and background final states is very useful as initial guidance for usage in experimental analyses. Their particular limitations are discussed.

Weaknesses

1 - The paper claims to limit itself to calorimeter information. At best the information used is motivated by calorimeter signal definitions, but with the used detector smearing codes (DELPHES) important features of a calorimeter are not considered (e.g. spatial/radial distribution of particle energy in detector and its effect on IR safety). Such signal features, together with possible problems introduced by the actual signal extraction and processing, can have important effects on the tagging performance.
2 - The study is limited to very specific (contained) top quark decays in a narrow phase space. This is fair and clearly stated, but it limits the validity of the study especially concerning application in experimental analyses.
3 - The use of pseudo-rapidity especially in taggers using trees or pair-wise combinations is not obvious, as some of the inputs may be massive (I assume the jet constituents are considered massless in the "calorimeter-signal-like" representation). Also, all FastJet clustering algorithms use rapidity.
4 - The description of the various taggers gets a bit technical in some cases and generates a relatively large number of - sometimes quite similar - acronyms as well (I assume it is somewhat unavoidable).
5 - Some of the captions describe the figures and tables very poorly (see suggestions in attached file).
4 - The general language is unconventional in parts and may be found inappropriate by some readers - and may degrade the messages of this paper. Also, references to citations are repeated quite often, which disturbs the readability as well.
5 - The abstract could be more meaningful in terms of providing some hint on the findings.

Report

The paper provides an important overview on state-of-the-art applications of machine learning techniques for tagging boosted top quark decays. A comparison of the performance of several taggers using jet images, jet kinematics and explicit guidance from physics modeling is provided for the signal efficiency and accuracy, and the background rejection. Signal and background final states are generated by Monte Carlo and used in a simple representation of the corresponding detector signals. All three approaches with their various machine-learning-based tagging technologies perform similar, with no indication of an outstanding deep network architecture.

The paper contains an important survey for further evaluation in colliding beam experiments and reflects a significant amount of original work. The studies are discussed with significant amounts of details concerning the machine learning configurations. The results are clearly summarized and their comprehensive presentation provides starting points for future applications in larger phase space and in realistic experimental environments. I consider this paper an important milestone in the certification of machine learning techniques for physics analysis, especially in the context of present and future collision environments at the Large Hadron Collider.

While the results are sound and well presented, the paper suffers from a rather short abstract which does not provide much information on why this paper is useful to read. It also has minor inconsistencies in nomenclature and a partly unconventional language. Some other mostly technical issues summarized in the attached file should be addressed as well. Once that is done, I am happy with recommending it for publication.

Requested changes

1 - Provide a more meaningful abstract with a few more details on the scope of the study and a hint on the findings. I have no further suggestions on changes to the content and results. Some technical suggestions follow.
2 - Read the complete manuscript in detail and remove all inconsistencies in nomenclature and references to improve readability.
3 - In the same spirit, please try to improve the figure and table captions.
4 - Please consider the comments in the attached file.

Attachment

  • validity: high
  • significance: top
  • originality: top
  • clarity: high
  • formatting: excellent
  • grammar: excellent

Author:  Tilman Plehn  on 2019-07-24  [id 572]

(in reply to Report 1 on 2019-05-13)

We are very grateful to all referees for carefully reading the manuscript, and we hope that we have accomodated all comments.

Report 3

  1. We have added a corresponding paragraph;
  2. We have made an attempt to unify the introductions, the relevant information should be included, except for the more sociologiacal aspect of how much effort was actually spent in optimizing each tagger;
  3. We added some discussion to the introductions of the methods and to Sec.4.

Report 2

  1. We describe some of the reasons in the text: the (new) LDA is based on an autoencoder setup, TopoDNN is simply not very optimized, and EFP removes information;
  2. The descriptions are more unified now, but we also leave the individual authors the decision which aspects of the tagger they want to emphasize. We deliberately did not include speed explicitely, because it would be hard to measure, but the number of parameters in Tab.1 should give a good estimate. Concerning overtraining, our test setup should catch that, and some of the tools have included for instance dropout (but we left the architectures to the author teams).

Report 1

  1. We expanded the abstract.
  2. We went through the paper carefully and tried to unified style, citations, spelling, etc.
  3. Almost all of the figure captions have been expanded, including the inspirational Fig.4;
  4. Thank you for the careful reading, we took care of all commments.

---

## Round 2 · Referee Report · Anonymous (Referee 2) · 2019-5-17

Strengths

1) Comparison of important tools and methods for future experimental analyses at the LHC.

2) State-of-the-art analysis setup, including a fast-detector simulation to mimic uncertainties on the inputs of the tagging methods.

3) The authors provide public data sets to allow for reproducibility and validation of their results and as a playground for future methods.

4) While many taggers have been included in the comparison, the authors have attempted to structure the comparison by grouping them in three categories, i.e. image-based taggers, 4-vector taggers and theory-inspired taggers.

5) Explicit mentioning of issues that need addressing before proliferation of the methods to experimental analysis should occur.

Weaknesses

1) Some of the methods used have been described fairly briefly and with varying level of detail.

2) Taggers from the 3 categories perform similarly. Thus, no intuition can be obtained with respect to the preferred inputs to the taggers, or even the architecture of the taggers. However, this can be counted as one outcome of the study.

3) Speed has not been included in the performance measure of the different methods.

Report

The authors compare the performance of different multivariate top taggers. Many of the taggers have been designed and published by subsets of the authors before. To allow for an objective comparison of the different methods, event samples are provided, including detector effects. The authors classify the taggers into three different groups: image-based, 4-vector-based and theory-inspired. The performance is shown using ROC curves. The authors find that tagging performances are broadly similar and do not depend on the categorisation of the taggers. Yet, some of the taggers show a much worse performance than others. It would be interesting to know if the authors have an intuition for why that is.
However, the authors clearly show that the tagging methods they proposed perform well and reliably, over a large signal efficiency range, in separating top quarks from QCD backgrounds.
I recommend publication in SciPost, but would like to ask the authors to consider the requests below.

Requested changes

1) Can the authors think of a reason why few taggers show a much worse performance than others? Stating such a reason explicitly in the text would add value to the paper.

2) Can the authors unify the way they present the different methods? Taggers have been described with a varying level of detail. For example, only for some (very few) of the taggers the issue of overtraining or speed has been discussed. As stated in the conclusions, the major difference between the taggers are not different inputs used, but merely the network structures. Thus, all of them could mention the time needed for training, the computational infrastructure used for training and the level of hyper-parameter optimisation and how they prevent and check for overtraining.

---

## Round 2 · Referee Report · Anonymous (Referee 3) · 2019-5-20

Strengths

  1. The paper has a broad assessment of NN taggers on a level playing field
  2. The paper uses state of the art techniques
  3. The paper limits itself to algorithms broadly applicable to the LHC community
  4. The paper is structured in a way to elucidate where physics information is most utilized
  5. There is a discussion of the limitations of the scope of this paper

Weaknesses

  1. The details of the samples are limited
  2. It is not clear whether the gains will translate into real gains on LHC data
  3. It is not clear what events are being missed by which algorithm (despite showing correlations)
  4. It is hard to follow the algorithm definition. Language is inconsistent.
  5. It would be nice to know about the architecture depth (Number of layers/iterations) in addition to just the number of weights

Report

This paper is a first comprehensive comparison of ML-based taggers using the same information and provided a systematic dataset for comparison. This work touches a number of active developments going on in the field of ML and jet substructure tagging. In particular, it elucidates the gain in ML and what choice of inputs and architectures can lead to maximal gains. The work is an important step in the direction of understanding the use of ML for jet substructure.

Despite this being an important step, there is still a lot of work that is needed to bring this study to the reality of the LHC. Also, it is a little difficult to follow all the details of the different NN architectures; though, I think this is just that each algorithm was presented in a different way. What is not clear from the paper is how this can actually impact the LHC. In particular, it would be nice to spell out in the introduction exactly what information is used and what is not. The document says Delphes is used with the ATLAS card taking only calorimeter information. I can go download the ATLAS card to figure out what is going, but I think it would be good to explicitly answer a few questions pertinent to the study. In particular, it is not clear :

  1. Whether there is any calo-clustering used

  1. In particular, how are particle overlaps taken into account

  1. Are Calo-level particle ids used in the corrections (ie is em-fraction). This is an effect that has come up in other NN studies.

  2. Do you correct for the eta and phi deviation of charged particles with respect to neutrals (not that these deviate substantially in the presence of a magnetic field)

  3. What are the pT thresholds of this cut (this is vaguely discussed in section 3.1)

  4. Also given the jets have |eta| < 2.0 that means jets go outside of the tracker volume this can lead to varying pT cuts vs eta. Is that taken into the simulation?

  5. What is the Pythia tune? How many parton legs are used?

  6. How is the jet selected (is it leading)

  7. Do you match the QCD jets to parton? or can selected jets be parton shower only? (is the QCD Matching based)

I am a little concerned that these effects might preferentially select one NN over the other. It is impossible to avoid any bias so I would add a table or paragraph that clearly states the assumptions. While I think the metrics used to draw conclusions on a whole are useful. In particular, I would like to know:

  1. How does each discriminator vary with respect to mass?

  2. How does each discriminator vary with respect to eta?

  3. if jets unmatched to hard partons are used, how does this discrimination vary.

For the QCD sample, we are talking about rejections of a factor of 1000 for about 1M events, which means that all these subtle details can change the performance quite a bit.

For the training, it was hard to find the level of optimization for each setup. In some cases like the Energy Flow Polynomials, it is clear. However, in some cases, it is not clear whether the number of parameters was scanned to find an optimum or not. Can a sentence be added to each section explicitly stating this? Also, it would be nice to know the number of layers in addition to parameters. This can affect things like general algorithm complexity.

Regarding the analysis of the performance, it is clear that some networks are better than others. The text is good at broadly classifying these differences. However, it is not clear whether these translate to meaningful effects in the collider environment. Have you :

  1. Looked at the variance of the minimum pT cut of the particles

1. Compared the performance trained on 1 tune vs another? (...)

If not, I eagerly await a follow-up paper where you start to address the robustness of these conclusions.

Some more smaller comments are below

I think "calorimeter image" is a misnomer since you are ignoring many physical effects that would make this an actual calorimeter. Particularly the fact that eta and phi positions at the calorimeter surface are different for charged and neutral particles due to propagation in a magnetic field. These are effects that are largely well understood, but ignored here and not the point of this paper. However, given angular resolution is such a critical component of this study, I am wondering if a discussion, later on, should be added.

Section 3 taggers

Small grammar error: If gave it the energy deposition It is not clear whether this pixelation of the jet image is on top of the clusters or not. Perhaps an addition to section 2 with definitions would help.

P-CNN: One of the distinguishing features of resnet is the ability to skip layers. I don't think this is present in this setup. From the description, this sounds just like a large CNN. Can this really be considered similar to resnet?

Figure 5 would benefit from a ratio plot

Figure 6 It looks like some of the variables like the nSub have very different distributions on their outputs. Did you consider a transformation to regularize the outputs? ( such as feeding the outputs through a small dense network)

Docutils System Messages

Requested changes

  1. A paragraph with the simulation details covering the detector with thresholds/resolutions as well as the generation details
  2. Unify the language across the different NN architectures. Explicitly talking about weights, optimization, and layers
  3. Discussion about any physical differences in the tagger how is mass used? Correlation to 3 pronginess. What other additional info can be pulled in?

---

## Editorial Decision

published